# Population-Attributable Fractions of Personal Comorbidities for Liver, Gallbladder, and Bile Duct Cancers

**DOI:** 10.3390/cancers15123092

**Published:** 2023-06-07

**Authors:** Kari Hemminki, Kristina Sundquist, Jan Sundquist, Asta Försti, Vaclav Liska, Akseli Hemminki, Xinjun Li

**Affiliations:** 1Biomedical Center, Faculty of Medicine and Biomedical Center in Pilsen, Charles University in Prague, 323 00 Pilsen, Czech Republic; 2Division of Cancer Epidemiology, German Cancer Research Center (DKFZ), Im Neuenheimer Feld 580, 69120 Heidelberg, Germany; 3Center for Primary Health Care Research, Lund University, 20502 Malmö, Sweden; kristina.sundquist@med.lu.se (K.S.); jan.sundquist@med.lu.se (J.S.); a.foersti@dkfz.de (A.F.); xinjun.li@med.lu.se (X.L.); 4Department of Family Medicine and Community Health, Icahn School of Medicine at Mount Sinai, New York, NY 10029-6574, USA; 5Center for Community-Based Healthcare Research and Education (CoHRE), Department of Functional Pathology, School of Medicine, Shimane University, Izumo-shi 693-8501, Japan; 6Hopp Children’s Cancer Center (KiTZ), 69120 Heidelberg, Germany; 7Division of Pediatric Neurooncology, German Cancer Research Center (DKFZ), German Cancer Consortium (DKTK), 69120 Heidelberg, Germany; 8Department of Surgery, University Hospital, School of Medicine in Pilsen, 323 00 Pilsen, Czech Republic; 9Cancer Gene Therapy Group, Translational Immunology Research Program, University of Helsinki, 00290 Helsinki, Finland; 10Comprehensive Cancer Center, Helsinki University Hospital, 00290 Helsinki, Finland

**Keywords:** hepatocellular carcinoma, comorbidity, risk factor, bile duct infection, alcohol, viral infection

## Abstract

**Simple Summary:**

Liver cancer is often used as a general term for cancers of the liver (hepatocellular carcinoma, HCC), the gallbladder, and the bile ducts. The well-known risk factors are alcohol and viral hepatitis, but these are risk factors of mainly HCC. For gallbladder cancer, gallstones are important risk factors, and for bile ducts, infections in the ducts are important. For all these cancers, autoimmune diseases and diabetes increase risk. This study shows that these risk factors, in combination, explain 50% or more of the causes of these cancers. The novelty of the present study was the use of national Swedish hospital records for potential risk factors (comorbidities) of hepatobiliary cancers and the estimation of subsequent risks of hepatobiliary cancers in these patients. The underlying mechanism for these cancers is a chronic infection which should be considered a marker of disease progression and a possible target for intervention.

**Abstract:**

Background: We aim to estimate population-attributable fractions (PAF) for 13 comorbidities potentially predisposing to hepatobiliary cancer of hepatocellular carcinoma (HCC), gallbladder cancer (GBC), cancers of the intrahepatic and extrahepatic bile ducts (ICC and ECC), and ampullary cancer. Methods: Patients were identified from the Swedish Inpatient Register from 1987 to 2018 and cancers from the Swedish Cancer Registry from 1997 through 2018. PAFs were calculated for each comorbidity-associated cancer using a cohort study design. Results: For male HCC, the major individual comorbidities (PAF > 10) were diabetes, alcohol-related liver disease, and hepatitis C virus infection. For female HCC, diabetes and autoimmune diseases were important contributors. For female GBC, gallstone disease was an overwhelming contributor, with a PAF of 30.57%, which was also important for men. The overall PAF for male ICC was almost two times higher than the female one. For ECC and ampullary cancer, infection of bile ducts was associated with the highest PAF. Conclusions: The 13 comorbidities accounted for 50% or more of the potential etiological pathways of each hepatobiliary cancer except female ICC. The underlying convergent mechanism for these cancers may be chronic inflammation lasting for decades and thus offering possibilities for intervention and disease monitoring.

## 1. Introduction 

Hepatobiliary cancers are a heterogeneous group of malignancies of the liver and the biliary tract, including hepatocellular carcinoma (HCC), gallbladder cancer (GBC), cancers of the intrahepatic and extrahepatic bile ducts (cholangiocarcinomas ICC and ECC), and ampullary cancer (cancer of the ampulla of Vater) [1,2,3,4]. The incidence of these cancer varies globally, which can be explained in part by regional risk factors, such as hepatitis B (HBV) or C virus (HCV) for HCC in Eastern Asia and Africa, liver flukes for ICC and ECC in Southeast Asia, and unknown risk factors for GBC in Western South America [1,2,3,4]. In Sweden, HCC incidence/mortality is higher in men compared to women, but the opposite is the case for GBC; for both cancers, an incidence peak was observed around the year 1980, ascribed to Thorotrast exposure, and for HCC, only a new increase in incidence started from the year 2000 onwards [5,6]. In developed countries, alcohol and HCV are important risk factors for HCC, with other, globally increasing lifestyle-related risk factors of obesity, low physical activity, type 2 diabetes, and non-alcoholic fatty liver disease (NAFLD) [1,2,3,4,7,8]. Autoimmune diseases and family history are additional risk factors [9,10]. GBC shares some risk factors with HCC, including obesity and family history, and features other unique risk factors, including gallstone disease, local bacterial infections, and biliary structural abnormalities [1,8,11]. ICC and ECC also share some risk factors with HCC, but biliary tract infections are the dominant risk factor, while associations with alcohol, HBV, and HCV are weak [1,8,12,13,14,15]. Survival in hepatobiliary cancers is poor; five-year survival in HCC, ECC, and GBC is reaching over 20%, is less than 10% for ICC, and over 30% for ampullary cancer [4,16,17,18]. The only curative treatment for these cancers is surgery, and for HCC, additional liver transplantation; for HCC sorafenib and GBC and bile duct cancers, in addition to chemotherapy, immunotherapies have become available [1,5,17,19].

For HCC, the convergent mechanism of action for the diverse risk factors is thought to be chronic inflammation, necroinflammation, and immune disturbance triggered by the inflammatory processes; this mechanism may account for 90% of HCC cases [3,20,21,22]. Data for GBC and biliary tract cancers similarly suggest that chronic inflammation is a shared mechanism that may underlie most of the cases [8].

The population impact of a risk factor for cancer depends on the magnitude of its relative risk and the proportion of the exposed population, which can be summarized as the population-attributable fraction (PAF) [23]. PAF (also known as population excess fraction) defines the proportion of cancer that can be attributed to the exposure (factor) under study [24]. Usually, the size of the exposed population (X) is not known and needs to be estimated. Similarly, estimates of the relative risk are derived from the literature based on historical exposure settings. Thus, much of the global literature on PAFs is based on estimated or extrapolated X and relative risk. This method was used in a UK analysis of PAF for all cancers, including liver cancer and GBC [25]. Male liver cancer (including intrahepatic bile ducts) was assigned a PAF of 24.1% for tobacco, 23.4% for overweight plus obesity, 10.2% for alcohol, and 10.1% for infections [25]. For women, the percentages were 13.0, 23.0, 1.4, and 8.5, respectively. Of note, these PAFs cannot be summed up, but when joint exposures were considered, 53.2% of male and 39.6% of female cases were explained by these risk factors. For GBC, only overweight plus obesity was considered, with PAF of 12.4% for men and 22.8% for women. A more direct way is to assign disease causation from the underlying liver diseases based on diagnostic criteria. This was done in the Swedish national HCC register [26]. The two most common estimated causes of HCC were HCV in 27% and alcohol in 23% of the cases [26]. The problem with this approach is that diagnostic signs are no proof of causation.

We present here a novel approach to assessing PAF with the help of nationwide hospital discharge data on 13 comorbidities that may predispose to hepatobiliary cancer. These 13 comorbidities define the population at risk, which is followed for hepatobiliary cancers. The actual patient numbers and their relative risks are used to derive PAF for each comorbidity. The goal is to provide an estimate of the contribution of the selected comorbidities and conditions to hepatobiliary cancers.

## 2. Materials and Methods

### 2.1. Study Design

The study was conducted as a cohort study.

### 2.2. Study Population

The participants were diagnosed with selected comorbidities that were known or assumed risk factors of hepatobiliary cancer.

### 2.3. Data Sources

The patients diagnosed with the comorbidities were obtained from the nationwide Swedish Inpatient Register starting in 1987, by which time the register had reached full national coverage. We were, however, not confined to the Inpatient Register as Sweden additionally has a nationwide Outpatient Register, operating since 2001, and a Primary Health Care Register in which diagnoses are collected from 21 of 22 regions in Sweden with an estimated coverage of 85%. Compared with the Inpatient Register, additional patients were found in these registers, but as we aimed at high diagnostic accuracy, we decided to use patients from the Inpatient Register only. Staff in hospitals are medical specialists, and medical care is available to the whole Swedish population, including admissions to hospitals. A total of 3.53 million persons out of a total of 13.6 million were diagnosed with any of the 13 comorbidities ensuring that we were able to cover diseases in the total population [8].

The comorbidities were identified using codes of the International Classification of Diseases (ICD) between 1987 and 1996 using ICD-9 and from 1997 onwards, ICD-10 to the end of the year 2018; the follow-up time was from 1997 to 2018 [8]. As no individual data are available for smoking and alcohol consumption, we used COPD and alcohol-related liver diseases as proxies. According to a Swedish survey on COPD, 89% of men and 64% of women were smokers, with PAFs of 64% and 29% [27]. For autoimmune diseases, liver-specific autoimmune hepatitis and primary biliary cirrhosis were analyzed separately, and all other 41 types of autoimmune diseases constituted ‘other autoimmune diseases’ as described elsewhere [28]. Persons were assigned to individual comorbidities based on their first diagnosis, which implied that no person was counted more than once. Cancer data were covered for the years 1997 through 2018 for the whole population of Sweden. Cancers were identified by the Swedish Cancer Registry using ICD-10 codes for primary HCC, C22.0, for ICC C22.1, for GBC C23.9, for ECC C24.0 (including Klatskin tumors), and for ampulla of Vater C24.1 [8]. The linkages between the different registers were done using the personal identification number assigned to each resident in Sweden and replaced by a serial number to preserve people’s integrity. 

### 2.4. Outcome Measures 

We published recently hepatobiliary cancer risks in persons hospitalized for 13 comorbidities [8]. In this paper, we used a standardized incidence ratio (SIR) as a measure of relative risk [8]. However, SIRs for different exposures are not comparable with each other because they are calculated using different exposed groups, and the numerical ranking of SIRs may be biased [29]. Thus, in the present study, we used the same 13 comorbidities and calculated the risks using a hazard ratio (HR). 

When PAFs are assessed based on estimated exposed groups, individual PAFs cannot be added up because a proportion of the population is exposed to multiple risk factors, and an adjustment is required [25]. However, here the case is different as no one is counted for more than a single exposure. The definition of considering the first comorbidity technically avoids the problem of multiple counting of comorbidities in the same patients. We investigated specifically the question of how commonly a patient was diagnosed with more than a single comorbidity among these 13 comorbidities. According to the Inpatient Register, 94.9–99.4% of the patients were diagnosed with the sole comorbidity. 

### 2.5. Statistical Analyses and Adjustment Variables

HRs were calculated for subsequent hepatobiliary cancer between each of the comorbidities and the background population without comorbidity from 1997 to 2018. HRs were adjusted for five-year age brackets, gender, period (five years group), highest educational level (as a proxy for socioeconomic status, which is associated with cancer risk), and geographical region. The 95% confidence interval (95% CI) of the HR was calculated assuming a Poisson distribution. 

The PAF for comorbidity i was calculated according to the formula: PAF_i_ = X_i_ × (HR_i_ − 1)/HR_i_, 
where X_i_ is the proportion of cancer patients with comorbidity i of all cancer patients, and HR_i_ is the relative risk of comorbidity i, i.e., Miettinen formula no. 9 [23]. Overall, PAF was the sum of individual comorbidity-associated PAFs. The 95% confidence intervals (95% CIs) were calculated according to Rothman and Greenland formula 16–25 considering variance for the proportion and the HR [29]. A previous paper on PAF calculation pointed out that the underlying data (calculation of HRs) should be adjusted for possible confounding factors [30]. This is exactly what we have done, as described in the previous paragraph. 

When PAFs for the five cancers were plotted by each comorbidity, nominal PAFs were used, and no adjustment was made for the size of the underlining population.

When HRs were discussed, only statistically significant associations (i.e., 95% CI did not include 1.00) were referred to.

## 3. Results

The present study covered 18,598 hepatobiliary cancers diagnosed between 1997 and 2018. Of these, HCC accounted for 46.6%, GBC for 22.8%, ICC for 12.3%, ECC for 11.4%, and ampullary cancer for 6.9%.

Male cancer risks for the 13 comorbidities are shown in Table 1. The overall HRs were increased for all cancer types, being highest for HCC (6.25), followed by ECC (5.47), and fairly equal for the other sites at over 4.00. A large proportion of the HRs were significant, and for ICC and HCC, all but one were significant. For 4/13 comorbidities (COPD, infection of bile ducts, diabetes, and other autoimmune diseases) conferred a risk at all subsites. Some comorbidities conferred very high risks (>10), which for HCC were HCV (67.42), HBV, hepatitis of other kinds, and primary biliary cirrhosis. For ICC, high associations were found in patients diagnosed with HBV, hepatitis of other kinds, and infections of bile ducts (56.35); infections of bile ducts also conferred high risks for GBC (33.83), ECC (108.51, highest in this study), and ampullary cancer (57.81). The total numbers of each cancer are shown in the footnote, as these numbers are the denominators in the calculation of exposed proportions.

Female associations are shown in Table 2. All overall associations were below the male ones, and particularly HRs for HCC and ICC were halved. Gallstone disease, infection of bile ducts, diabetes, and other autoimmune disease conferred a risk for all cancers. Primary biliary cirrhosis was more common in women compared to men, and significant associations were also noted for ICC and ECC. In general, the female risk profile resembled that of men, but curiously, obesity was associated with no female cancers. 

The data from Table 1 (proportion of exposed and HR) was used to derive male PAF estimates (Table 3). The combined male PAF was 68.71% for HCC, with a major contribution by diabetes (17.80%), alcohol (12.93%), and HCV (11.75%). The overall PAF for ICC was 57.44%, with contributions by individual comorbidities of other autoimmune diseases (17.02%), infection of bile ducts (14.49%), and diabetes (12.31%). The overall PAF for GBC was 56.86%, with a prominent share of gallstone disease (23.87%) seconded by infection of bile ducts (11.46%). For ECC, the overall PAF was 61.45%, with infection of bile ducts (25.04%, the highest individual male PAF) and diabetes (11.04%) as the prominent contributors. For ampullary cancer, with an overall PAF of 52.25%, three comorbidities dominated, infection of bile ducts (16.89%), gallstone disease (10.90%), and diabetes (11.63%). 

Male PAF data for each comorbidity are summarized in Figure 1. PAFs for alcohol-related liver disease, all kinds of hepatitis, liver-specific autoimmune diseases, and diabetes dominated HCC. Gallstone disease showed a prominent PAF for GBC, and infection of bile ducts dominated ECC. The aggregation of many cancers with high PAFs around gallstone disease, infection of bile ducts, diabetes, and other autoimmune diseases shows the overall weight of these comorbidities. 

In Figure 2, we collected male PAFs for each comorbidity, illustrating the differential shares of comorbidities as risk factors of specific hepatobiliary cancer. HCC was the dominant cancer in patients with alcohol-related liver diseases, viral hepatitis, and primary biliary cirrhosis. Autoimmune hepatitis was associated with equal PAFs for HCC and ICC. Infection of bile ducts conferred high PAFs in all parts of the duct cancers, with prominence to the distal parts. Gallstone disease targeted GBC locally. Diabetes and other autoimmune diseases shared PAF for all cancers. 

For women, the combined PAFs were for HCC 52.50%, ICC 29.06%, GBC 48.18%, ECC 57.24, and ampullary cancer 48.39% (Table 4). For female HCC, diabetes (10.78%) and other autoimmune diseases (10.01%) were important contributors; PAF for alcohol-related liver disease was only 4.57%. For ICC, infection of bile ducts (8.67%) and other autoimmune diseases (8.39%) were the most prevalent comorbidities. For GBC, gallstone disease (PAF of 30.57%, the highest female PAF) accounted for 2/3 of all PAF. Female ECC had a prominent contribution by infection of bile ducts (23.56%) and gallstone disease (12.64%). For ampullary cancer, infection of bile ducts (16.67%) and gallstone disease (12.94%) were the main contributors.

Female PAF data for each comorbidity are summarized in Figure 3. One can visualize the small PAFs of alcohol-related liver disease and viral hepatitis compared to the prominent PAFs for gallstone disease and infections of bile ducts. The four prominent comorbidities were the same as in men: gallstone disease, infection of bile ducts, diabetes, and other autoimmune diseases. 

In Figure 4, female PAFs are shown for each comorbidity, illustrating the differential shares of comorbidities as risk factors for specific hepatobiliary cancers. There is a difference to Figure 2 for autoimmune hepatitis because of the small cancer numbers. Some true differences are seen for a larger share of HCC in patients diagnosed with hepatitis of other kinds and NAFLD. 

## 4. Discussion

The novelty of the present approach was our use of medically diagnosed comorbidity data to assess the risks of specific types of hepatobiliary cancer in the same individuals to derive the related PAFs. As we measured cancer risk directly in the same individuals, no extrapolations were needed, offering a significant advantage over the traditional epidemiological approach where both the size of the exposed populations and their cancer risk are extrapolated [25]. The combined male PAFs were 68.71% for HCC, for ICC 57.44%, for GBC 56.86%, for ECC 61.45%, and for ampullary cancer 52.25%. For women, the combined PAFs were for HCC 52.50%, for ICC 29.06%, for GBC 48.18%, for ECC 57.24, and for ampullary cancer 48.39%.

The novelty is reflected by the fact that no similar data are available in the literature. Papers on PAFs for environmental risk factors do not consider medical conditions [25,31]. For infections, PAF estimates have been published, and several papers have provided PAF estimates for HCC [32,33]. Globally, 44% of HCC cases were attributed to HBV infection and 21% to HCV infection, with the majority of cases occurring in developing countries [32]. Lifestyle risk factors, including alcohol drinking and obesity, were important attributable causes in developed countries [32].

We did not consider family history a risk factor even though data were available [9]. However, family history was covered partly indirectly: many of the present comorbidities in one family member are associated with HCC in another member; for example, if a family member was diagnosed with alcohol-related liver disease or HBV infection, another family member had an increased risk of HCC [9]. Similarly, diabetes was associated with ampullary cancer in a family member [9].

Because of the definition of PAF (proportion × risk), the most prominent comorbidities had to be common and confer a high HR. These were met by diabetes for HCC, other autoimmune diseases for ICC, gallstone disease for GBC, and infection of bile ducts for both ECC and ampullary cancer. An important observation, reminding of the direction of bile flow, was the gradient of PAFs in gallstone disease patients, which were highest for local GBC, followed by ECC and ampullary cancer, and at a lower level by ICC; gallstone disease did not influence HCC. The prominent comorbidities were sex-concordant. One can speculate that adding the unaccounted and non-symptomatic patients would boost the PAFs to the extent that the majority of the disease associations could be explained by the present comorbidities. A word of caution should remind us of surveillance bias which is almost always present when persons undergo medical examination [34]. However, this kind of confounding may be small in aggressive cancers, such as hepatobiliary cancers. 

A PAF estimate for HCC has been published for the USA, and the ranking was: metabolic disorders, i.e., mainly type 2 diabetes (32%), HCV (20.5%), alcohol (13.4%), smoking (9%), HBV (4.3%), and genetic disorders (1.5%); the overall PAF was 59.5% [33]. Although the overall PAF was somewhat lower than our present one, all individual PAFs were higher than ours, although alcohol only marginally.

As shown above, the overall PAFs were lower for female cancers compared to male cancers. The largest differences were for HCC 52.50% vs. 68.71% and particularly for ICC 29.06% vs. 52.50%. The sex difference for HCC depended on much higher male contributions by alcohol, HCV, HBV, and diabetes. These comorbidities also contributed to male excess of PAF for ICC, but additionally, infections of bile ducts and autoimmune diseases were more common in men. One may also note that in women, obesity was not a risk factor for any of the hepatobiliary cancers. 

Hepatobiliary cancers are fatal cancers with five-year survival ranging from about 10 to 30%, which shows that there is an unmet need to control these cancers through surveillance and risk prevention [4,16,17,18]. At one level, the focus of improvements may be on avoidance or reduction of exposure to risk factors, such as smoking, alcohol consumption, overweight, physical inactivity, and prevention and treatment of viral hepatitis [35]. Screening options may be feasible for cirrhotic patients diagnosed with high-risk liver autoimmune diseases [36]. The other focus could try to benefit from the mechanistic understanding of the underlying chronic infection and related immune disturbance [3,8,20,21,22]. As a decades-long chronic infection is a common carcinogenic mechanism for these hepatobiliary cancers, tackling early phases of the disease process may offer new therapeutic options and pathways that allow monitoring of disease progression from its preclinical phase [22]. Good medical control of diabetes and autoimmune disease and intervention in bile duct infections and gallstone disease may reduce cancer risk [37]. Aspirin reduces the risk of HCC [38]; aspirin, other non-steroidal anti-inflammatory drugs, and statins have been reported to reduce risks of HCC, GBC, and CCA [39,40,41].

### Strengths and Limitations of the Study

The advantages of this approach are that the exposed populations are defined (mostly through medical specialists) rather than estimated, and the relative risks for cancer are measured rather than extrapolated. Another advantage was that the comorbidity integrated the influence of any known and unknown risk factors shared by it and hepatobiliary cancer. A further advantage was that there was no need to consider interactions because these were part of the patient’s pathophysiology. The convention of only considering the patient’s first comorbidity simplified the inclusion criteria but could have skewed comorbidities toward early-onset diseases. However, as we showed in Methods, it was very rare that a person was diagnosed with more than one comorbidity. These last two points justified summing up the PAFs to a joint PAF. 

The disadvantage of the approach was that we were not able to cover all exposed individuals because they did not receive medical treatment (e.g., most alcohol consumers, only the extremely obese individuals, and smokers), no codes were available for comorbidity (e.g., primary sclerosing cholangitis) or because follow-up time was limited. It is known that cancers caused by chronic stimulation and cirrhosis take decades to transform into cancer; for example, in HCV, a Swedish study showed that the maximal HCC risk was reached after 40 years of infection [42]. These coverage issues were discussed in more detail in our previous paper [8]. A further limitation is that the results may apply only to the Swedish population or those with similar exposure and risk relationships, such as in many other Western countries. With the above limitations and caveats in mind, it is remarkable that the male comorbidities explain between 68.71% of HCC and 52.25% of ampullary cancer etiology or etiological pathways.

## 5. Conclusions

The selected 13 comorbidities in patients diagnosed from the National Inpatient Register conferred high combined PAFs, of which the highest for men was 68.71% for HCC and women 57.24% for ECC. Considering the unaccounted patients with these comorbidities, it is likely that these 13 comorbidities account for the majority of the etiology or etiological pathways of these cancers. Alcohol-related liver disease and HCV were important comorbidities for HCC, but for the other cancers, gallstone disease and infection of bile ducts were most important; diabetes and other autoimmune diseases conferred large PAFs for all hepatobiliary cancers.

## Figures and Tables

**Figure 1 cancers-15-03092-f001:**
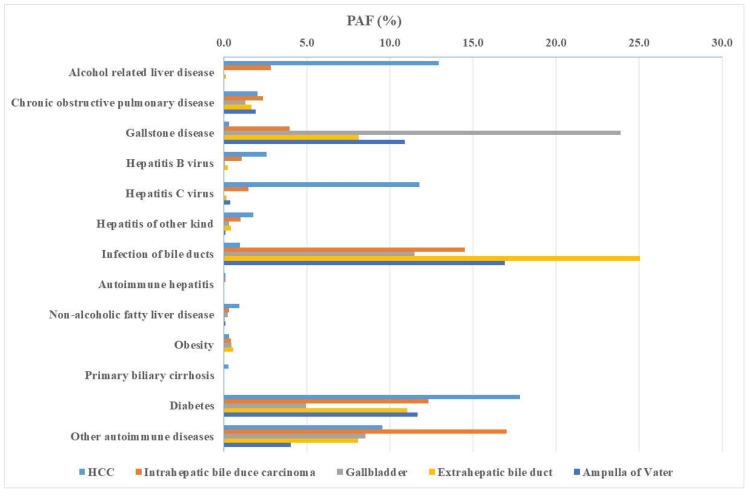
Male population attributable fraction (PAF) of individual hepatobiliary cancers after comorbidities. The figure summarizes data in Table 3 for each comorbidity.

**Figure 2 cancers-15-03092-f002:**
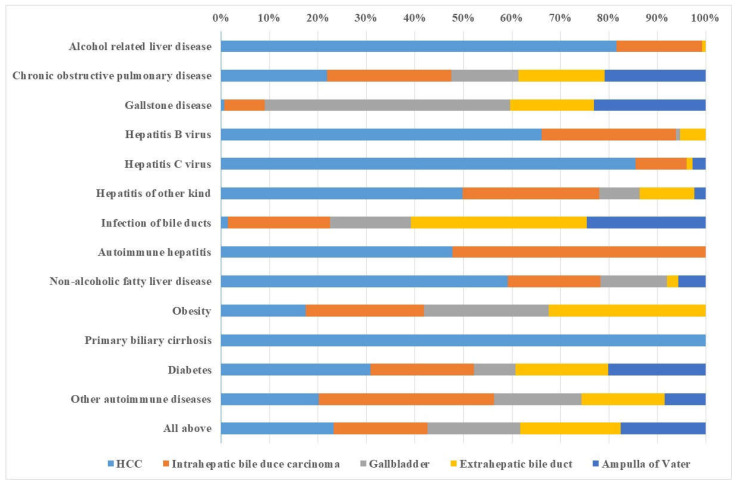
Male distribution of population-attributable fraction (PAF) of the liver, gallbladder, and bile duct cancers for each comorbidity. The figure summarizes data in Table 3 by considering the relative contribution of each hepatobiliary cancer for which the total is set at 100%.

**Figure 3 cancers-15-03092-f003:**
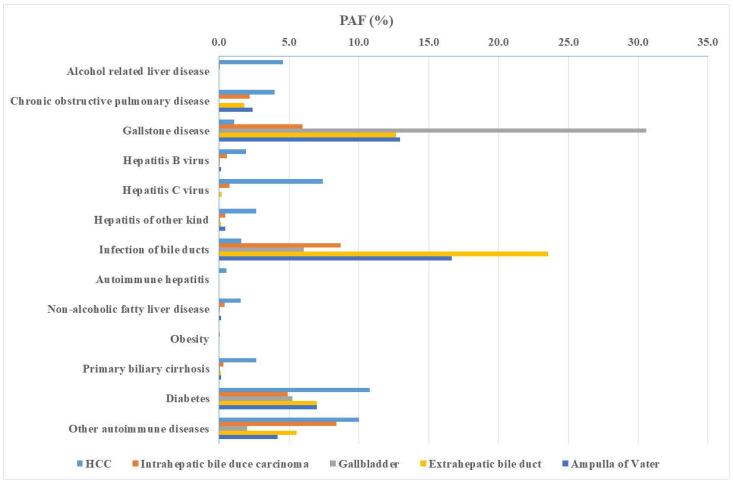
Female population-attributable fraction (PAF) of individual hepatobiliary cancers after comorbidities. The figure summarizes data in Table 4 for each comorbidity.

**Figure 4 cancers-15-03092-f004:**
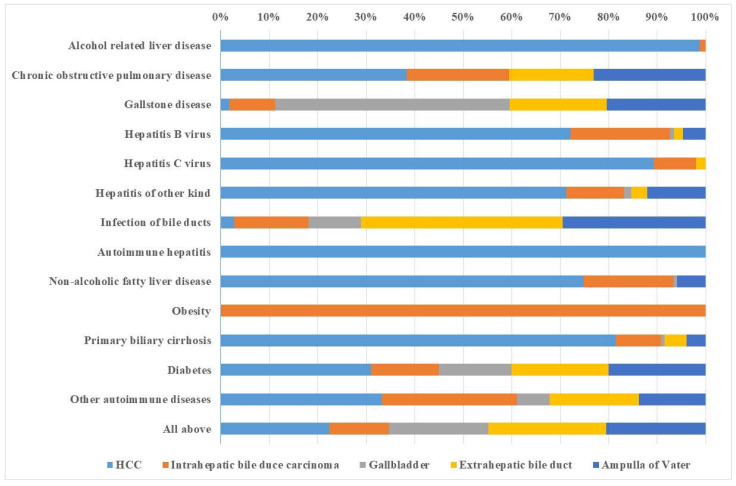
Female distribution of population-attributable fraction (PAF) of the liver, gallbladder, and bile duct cancers for each comorbidity. The figure summarizes data in Table 4 by considering the relative contribution of each hepatobiliary cancer for which the total is set at 100%.

**Table 1 cancers-15-03092-t001:** Subsequent risk of hepatobiliary cancer after comorbidities in men, 1997–2018.

	HCC				Intrahepatic Bile Duct Carcer				GBC				Extrahepatic Bile Duct Cancer				Ampullary Cancer			
Diagnosis of Comorbidities	O	HR	95% CI		O	HR	95% CI		O	HR	95% CI		O	HR	95% CI		O	HR	95% CI	
Alcohol-related LD	943	**6.70**	**6.24**	**7.19**	63	**2.02**	**1.56**	**2.61**	24	0.87	0.58	1.31	30	1.04	0.73	1.50	17	0.96	0.59	1.55
COPD	323	**1.63**	**1.45**	**1.83**	57	**1.88**	**1.43**	**2.48**	57	**1.34**	**1.02**	**1.75**	48	**1.59**	**1.19**	**2.14**	39	**1.53**	**1.09**	**2.16**
Gallstone disease	317	1.06	0.95	1.19	95	**1.89**	**1.53**	**2.34**	317	**7.17**	**6.29**	**8.17**	136	**2.85**	**2.38**	**3.41**	108	**3.45**	**2.81**	**4.25**
Hepatitis B virus	163	**39.27**	**33.57**	**45.95**	13	**14.78**	**8.55**	**25.56**	1	1.55	0.22	10.91	3	**3.99**	**1.28**	**12.38**	0			
Hepatitis C virus	740	**67.42**	**62.25**	**73.02**	19	**7.34**	**4.65**	**11.56**	2	0.99	0.03	3.97	4	1.77	0.66	7.71	4	**2.96**	**1.11**	**7.93**
Hepatitis of other kinds	113	**31.19**	**25.89**	**37.58**	12	**17.13**	**9.69**	**30.26**	4	**6.25**	**2.34**	**16.66**	5	**7.82**	**3.26**	**18.80**	1	2.44	0.34	17.34
Infection of bile ducts	81	**3.72**	**2.98**	**4.63**	167	**56.35**	**47.64**	**66.65**	135	**33.83**	**28.21**	**40.58**	275	**108.51**	**94.22**	**124.97**	121	**57.81**	**47.34**	**70.60**
Autoimmune hepatitis	5	**8.75**	**3.64**	**21.01**	1	**8.61**	**1.21**	**61.20**	0				0				0			
NAFLD	60	**17.63**	**13.67**	**22.74**	4	**6.00**	**2.25**	**16.02**	3	**5.17**	**1.66**	**16.05**	1	1.64	0.23	11.66	1	2.64	0.37	18.73
Obesity	65	**1.39**	**1.09**	**1.78**	14	1.50	0.89	2.55	12	1.71	0.97	3.02	14	**1.74**	**1.03**	**2.96**	2	0.42	0.11	1.68
Primary biliary cirrhosis	17	**33.42**	**20.46**	**54.60**	0				0				0				0			
Diabetes	1357	**5.39**	**5.05**	**5.74**	180	**4.42**	**3.74**	**5.23**	122	**1.86**	**1.53**	**2.26**	163	**3.80**	**3.19**	**4.53**	115	**3.47**	**2.82**	**4.28**
Other AID	893	**2.95**	**2.74**	**3.18**	240	**5.07**	**4.35**	**5.91**	166	**2.42**	**2.02**	**2.88**	139	**2.72**	**2.25**	**3.29**	66	**1.75**	**1.35**	**2.28**
All of above	5077	**6.25**	**5.91**	**6.61**	865	**4.03**	**3.57**	**4.55**	843	**4.37**	**3.86**	**4.94**	818	**5.47**	**4.80**	**6.24**	474	**4.47**	**3.81**	**5.24**

HCC = Hepatocellular carcinoma; GBC= gallbladder cancer; O = Observed; HR = Hazards ratio; CI = Confidence intervals; LD = Liver disease; COPD = Chronic obstructive pulmonary disease; NAFLD = Non-alcoholic fatty liver disease; AID = autoimmune disease. Bold types: 95% CI does not include 1.00. There were a total of 6207 HCC, 1132 intrahepatic bile duct carcinoma, 1143 gallbladder, 1088 extrahepatic bile duct, and 704 ampullae of Vater cancers.

**Table 2 cancers-15-03092-t002:** Subsequent risk of hepatobiliary cancer after comorbidities in women, 1997–2018.

	HCC				Intrahepatic Bile Duct Carcer				GBC				Extrahepatic Bile Duct Cancer				Ampullary Cancer			
Diagnosis of Comorbidities	O	HR	95% CI		O	HR	95% CI		O	HR	95% CI		O	HR	95% CI		O	HR	95% CI	
Alcohol-related LD	137	**5.65**	**4.74**	**6.73**	15	1.05	0.63	1.75	19	0.65	0.41	1.02	7	0.62	0.30	1.31	4	0.66	0.25	1.77
COPD	190	**2.06**	**1.77**	**2.40**	63	**1.67**	**1.28**	**2.18**	128	0.99	0.83	1.19	54	**1.52**	**1.14**	**2.02**	35	**1.64**	**1.15**	**2.34**
Gallstone disease	195	**1.16**	**1.00**	**1.34**	146	**1.89**	**1.59**	**2.25**	1097	**7.36**	**6.83**	**7.92**	193	**3.06**	**2.61**	**3.57**	109	**3.09**	**2.51**	**3.81**
Hepatitis B virus	49	**42.69**	**32.07**	**56.83**	7	**10.64**	**5.06**	**22.38**	2	1.59	0.40	6.32	1	1.99	0.28	14.08	1	3.69	0.52	26.24
Hepatitis C virus	185	**71.59**	**61.39**	**83.48**	10	**6.13**	**3.29**	**11.44**	1	0.31	0.04	2.18	3	2.35	0.76	7.31	0			
Hepatitis of other kinds	67	**37.79**	**29.64**	**48.19**	6	**6.67**	**3.00**	**14.87**	4	1.76	0.66	4.68	2	2.58	0.65	10.34	3	**6.91**	**2.23**	**21.44**
Infection of bile ducts	49	**4.76**	**3.59**	**6.32**	104	**25.85**	**21.09**	**31.67**	201	**15.23**	**13.20**	**17.59**	245	**80.60**	**69.60**	**93.34**	97	**49.25**	**39.47**	**61.45**
Autoimmune hepatitis	13	**20.63**	**11.96**	**35.57**	0				0				0				0			
NAFLD	39	**31.53**	**22.98**	**43.27**	5	**7.75**	**3.22**	**18.64**	2	1.30	0.33	5.19	0				1	3.30	0.46	23.47
Obesity	23	0.76	0.50	1.15	17	1.01	0.63	1.64	33	0.99	0.70	1.40	9	0.69	0.36	1.34	3	0.43	0.14	1.34
Primary biliary cirrhosis	67	**54.08**	**41.83**	**69.92**	4	**8.42**	**3.15**	**22.45**	3	1.29	0.32	5.18	2	**4.48**	**1.12**	**17.93**	1	3.82	0.54	27.17
Diabetes	362	**3.77**	**3.35**	**4.24**	96	**2.43**	**1.94**	**3.04**	276	**2.43**	**1.94**	**3.04**	109	**2.93**	**2.38**	**3.62**	61	**2.88**	**2.19**	**3.78**
Other AID	413	**2.49**	**2.23**	**2.78**	172	**2.29**	**1.92**	**2.72**	310	**1.25**	**1.11**	**1.41**	126	**1.83**	**1.50**	**2.22**	64	**1.59**	**1.22**	**2.08**
All of above	1789	**3.62**	**3.34**	**3.93**	645	**2.08**	**1.85**	**2.34**	2076	**3.57**	**3.32**	**3.84**	751	**4.60**	**4.03**	**5.26**	379	**3.67**	**3.09**	**4.37**

HCC = Hepatocellular carcinoma; GBC= gallbladder cancer; O = Observed; HR = Hazards ratio; CI = Confidence intervals; LD = Liver disease; COPD = Chronic obstructive pulmonary disease; NAFLD = Non-alcoholic fatty liver disease; AID = autoimmune disease. Bold types: 95% CI does not include 1.00. There were a total of 2467 HCC, 1153 intrahepatic bile duct carcinoma, 3101 gallbladder, 1027 extrahepatic bile duct, and 576 ampullae of Vater cancers.

**Table 3 cancers-15-03092-t003:** Population attributable fraction (PAF, %) of hepatobiliary cancer in men after comorbidities, 1997–2018.

	HCC			Intrahepatic Bile Duct Cancer			GBC			Extrahepatic Bile Duct			Cancer Ampullary Cancer		
Diagnosis of Comorbidities	PAF	95% CI		PAF	95% CI		PAF	95% CI		PAF	95% CI		PAF	95% CI	
Alcohol-related LD	12.93	11.51	14.35	2.81	0.46	5.16	0.00			0.11	0.00	2.37	0.00		
COPD	2.01	0.39	3.64	2.36	0.05	4.66	1.27	0.00	4.16	1.64	0.00	3.98	1.92	0.00	5.21
Gallstone disease	0.29	0.00	2.58	3.96	0.64	7.28	23.87	19.50	28.24	8.11	4.59	11.63	10.90	6.55	15.25
Hepatitis B virus	2.56	2.15	2.97	1.07	0.44	1.70	0.03	0.00	0.20	0.21	0.00	0.53			
Hepatitis C virus	11.75	10.83	12.66	1.45	0.66	2.23	0.00			0.16	0.00	0.57	0.38	0.00	0.97
Hepatitis of other kinds	1.76	1.42	2.10	1.00	0.39	1.60	0.29	0.00	0.64	0.40	0.00	0.81	0.08	0.00	0.36
Infection of bile ducts	0.95	0.59	1.31	14.49	12.06	16.92	11.46	9.33	13.60	25.04	21.58	28.50	16.89	13.52	20.26
Autoimmune hepatitis	0.07	0.00	0.14	0.08	0.00	0.25									
NAFLD	0.91	0.66	1.16	0.29	0.00	0.65	0.21	0.00	0.51	0.04	0.00	0.22	0.09	0.00	0.37
Obesity	0.29	0.00	0.85	0.41	0.00	1.38	0.44	0.00	1.25	0.55	0.00	1.49	0.00		
Primary biliary cirrhosis	0.27	0.14	0.40												
Diabetes	17.80	15.64	19.97	12.31	8.91	15.70	4.94	0.87	9.01	11.04	7.52	14.56	11.63	7.03	16.23
Autoimmune diseases	9.51	7.25	11.78	17.02	12.98	21.05	8.51	4.08	12.93	8.08	4.34	11.83	4.03	0.00	8.47
All of above	68.71	67.96	69.41	57.44	54.99	59.62	56.86	54.66	58.81	61.45	41.24	81.66	52.25	32.24	72.27

PAF = Population attributable fraction; HCC = Hepatocellular carcinoma; GBC=gallbladder cancer; CI = Confidence intervals; LD = Liver disease; COPD = Chronic obstructive pulmonary disease; NAFLD = Non-alcoholic fatty liver disease.

**Table 4 cancers-15-03092-t004:** Population-attributable fraction (PAF, %) of hepatobiliary cancer in women after comorbidities, 1997–2018.

	HCC			Intrahepatic Bile Duct Cancer			GBC			Extrahepatic Bile Duct Cancer			Ampullary Cancer		
Diagnosis of Comorbidities	PAF	95% CI		PAF	95% CI		PAF	95% CI		PAF	95% CI		PAF	95% CI	
Alcohol-related LD	4.57	3.45	5.69	0.06	0.00	1.32	0.00			0.00			0.00		
COPD	3.96	1.62	6.30	2.19	0.00	4.84	0.00			1.80	0.00	4.65	2.40	0.00	5.99
Gallstone disease	1.07	0.00	4.81	5.97	1.43	10.50	30.57	26.90	34.24	12.64	7.88	17.41	12.94	7.07	18.81
Hepatitis B virus	1.94	1.38	2.50	0.55	0.10	1.00	0.02	0.00	0.12	0.05	0.00	0.24	0.13	0.00	0.47
Hepatitis C virus	7.39	6.27	8.52	0.73	0.17	1.28	0.00			0.17	0.00	0.52			
Hepatitis of other kinds	2.64	1.98	3.31	0.44	0.02	0.87	0.06	0.00	0.20	0.12	0.00	0.40	0.45	0.00	1.05
Infection of bile ducts	1.57	0.94	2.20	8.67	6.84	10.50	6.06	5.10	7.01	23.56	20.13	26.99	16.67	12.95	20.40
Autoimmune hepatitis	0.50	0.21	0.79												
NAFLD	1.53	1.03	2.03	0.38	0.00	0.76	0.01	0.00	0.11				0.12	0.00	0.47
Obesity	0.00			0.02	0.00	1.44	0.00			0.00			0.00		
Primary biliary cirrhosis	2.67	2.01	3.33	0.31	0.00	0.65	0.02	0.00	0.16	0.15	0.00	0.42	0.13	0.00	0.47
Diabetes	10.78	8.17	13.39	4.89	2.03	7.75	5.23	2.48	7.98	7.00	3.83	10.16	6.98	3.05	10.92
Autoimmune diseases	10.01	6.20	13.83	8.39	3.71	13.07	2.02	0.00	6.04	5.55	0.78	10.33	4.18	0.00	9.95
All of above	52.50	50.78	54.08	29.06	25.69	32.07	48.18	46.75	49.51	57.24	36.32	78.16	48.39	25.68	71.10

PAF = Population-attributable fraction; HCC = Hepatocellular carcinoma; GBC= gallbladder cancer: CI = Confidence intervals; LD = Liver disease; COPD = Chronic obstructive pulmonary disease; NAFLD = Non-alcoholic fatty liver disease.

## Data Availability

All data generated or analyzed during this study are included in this published article.

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
