# Peer review of "Population-Attributable Fractions of Personal Comorbidities for Liver, Gallbladder, and Bile Duct Cancers"

_cancers, 2023, doi:10.3390/cancers15123092_

Round 1

Reviewer 1 Report

Very interesting and well written paper. I have the following comments:

1) Do the authors have some data also on the impact of baseline pharmacological therapies on these cancers? For example on the use of statins? In this regard cite the recent paper PMID: 32260179)

2) THe authors should comment more on the potential impact of diabetes on these cancers, given the high prevalence of this disease. in this regard cite the review PMID: 23845075 )

3) The figures are quite unclear. Explain them better in the caption

Author Response

Reply to Reviewer 1

Very interesting and well written paper. I have the following comments:

1) Do the authors have some data also on the impact of baseline pharmacological therapies on these cancers? For example on the use of statins? In this regard cite the recent paper PMID: 32260179)

Text was added to line 103-5, statins in 309-10, new ref is No 41.

2) THe authors should comment more on the potential impact of diabetes on these cancers, given the high prevalence of this disease. in this regard cite the review PMID: 23845075 )

Diabetes is discussed e.g,, line 283 ande 316-8. New reference is no. 37.

3) The figures are quite unclear. Explain them better in the captio

All corrected.

We thank the reviewer!

Reviewer 2 Report

The manuscript (ID: cancers-2439749) aimed to estimate population attributable fractions for 13 personal comorbidities potentially predisposing to liver, gallbladder and bile duct cancers.  

In this paper, it is necessary to enter several corrections (major revision).  

Some of the comments are:      

  • Line 41: As part of `Methods`, specify the study design applied in this paper.   
  • Line 53: Correct keywords, in accordance with the most important results presented in this paper (for example, a proxy was used for `smoking`, etc).  
  • Line 77: In addition to the data mentioned in the previous text, add a new paragraph in which the data on the incidence / mortality of hepatobiliary cancers in Sweden will be listed, citing the appropriate references.
  • Lines 94-95: Provide the appropriate reference for the claim presented in this sentence.
  • Lines 101-103: Move this sentence to the end of the Discussion section, where `Strength of the study` (and limitations) should be stated.
  • Line 103: Specify the goal of this paper at the end of the Introduction section. Determine whether the aim of this work is what the authors stated on Lines 38-41, as follows: `We aim to estimate population attributable fractions (PAF) for 13 comorbidities potentially predisposing to hepatobiliary cancer of hepatocellular carcinoma (HCC), gallbladder cancer (GBC), cancers of the intrahepatic and extrahepatic bile ducts (ICC and ECC) and ampullary cancer.`? Or is the goal of this work what is stated on Lines 97-98, as follows `We present here a novel approach of assessing PAF with the help of nationwide hospital discharge data on 13 comorbidities which may predispose to hepatobiliary cancer.`?  
  • Line 104: Either because they are missing or for better visibility and transparency, it is mandatory to introduce the missing subsections in the section on work methodology and/or systematize the presented data in an appropriate way, as follows: `Study design`, `Study population`, `Data sources`, `Measures`, `Variables`, `Statistical Analysis`.
  • Line 253: The Discussion section as a whole does not correspond to the objectives of the work stated in the Abstract or the title of this paper. If only the aim of the paper had been stated in the Abstract or in the title of the paper that the characteristics of `a novel approach of assessing PAF` would be presented, the discussion would have been satisfactory.
  • Lines 253-309: Reconstruct the Discussion section completely, as follows:
    • Ensure a logical flow and comprehensiveness in the presentation of the comparison of own results with the results of similar studies by other authors (not only own previous studies) in other countries, and
    • provide an adequate explanation for possible differences in results between different studies.
    • Discuss primarily PAF for the 13 comorbidities presented in this paper.
    • Give an explanation why variables that are not presented in the Results section of this paper are discussed, such as family history, aspirin, etc.
    • At the end of the Discussion section, introduce the subsection `Strength and limitations of the study`, with a discussion of all limitations, potential sources of biases, etc.
    • Pay special attention to the fact that in this version of the paper, the Discussion section ends with the citation of reference No. 30, and that a total of 34 references are listed in the list of References (references 31-34 are not cited in the text). Correct this.
  • Line 309: Discuss the differences in PAF (%) of hepatobiliary cancer after comorbidities, 1997-2018, by gender, in accordance with the results presented in this paper. Provide an explanation for the differences found.
  • Line 309: Discuss the differences in the distribution of population attributable fraction PAF of liver, gallbladder and bile ducts cancers for each comorbidity, by gender, in accordance with the results presented in this paper. Provide an explanation for the differences found.
  • Line 309: Especially discuss the result of this paper that the female risk profile showed that obesity was associated with no female cancers.  
  • Line 310: The Conclusions section correctly highlights the most important results presented in this paper, except for the sentence on Lines 318-321, which does not belong in the Conclusions section but in the Discussion section. Correct this.   

The quality of English language is appropriate.  

Author Response

Reply to Reviewer 2

The manuscript (ID: cancers-2439749) aimed to estimate population attributable fractions for 13 personal comorbidities potentially predisposing to liver, gallbladder and bile duct cancers.  

In this paper, it is necessary to enter several corrections (major revision).  

Some of the comments are:      

 Our reply below indicates lines for each comment.

  • Line 41: As part of `Methods`, specify the study design applied in this paper.   
  • 58
  • Line 53: Correct keywords, in accordance with the most important results presented in this paper (for example, a proxy was used for `smoking`, etc).  
  • 34
  • Line 77: In addition to the data mentioned in the previous text, add a new paragraph in which the data on the incidence / mortality of hepatobiliary cancers in Sweden will be listed, citing the appropriate references.
  • 91-4
  • Lines 94-95: Provide the appropriate reference for the claim presented in this sentence.
  • 126
  • Lines 101-103: Move this sentence to the end of the Discussion section, where `Strength of the study` (and limitations) should be stated.
  • 324-6
  • Line 103: Specify the goal of this paper at the end of the Introduction section. Determine whether the aim of this work is what the authors stated on Lines 38-41, as follows: `We aim to estimate population attributable fractions (PAF) for 13 comorbidities potentially predisposing to hepatobiliary cancer of hepatocellular carcinoma (HCC), gallbladder cancer (GBC), cancers of the intrahepatic and extrahepatic bile ducts (ICC and ECC) and ampullary cancer.`? Or is the goal of this work what is stated on Lines 97-98, as follows `We present here a novel approach of assessing PAF with the help of nationwide hospital discharge data on 13 comorbidities which may predispose to hepatobiliary cancer.`?  
  • 131
  • Line 104: Either because they are missing or for better visibility and transparency, it is mandatory to introduce the missing subsections in the section on work methodology and/or systematize the presented data in an appropriate way, as follows: `Study design`, `Study population`, `Data sources`, `Measures`, `Variables`, `Statistical Analysis`.
  • 135-179
  • Line 253: The Discussion section as a whole does not correspond to the objectives of the work stated in the Abstract or the title of this paper. If only the aim of the paper had been stated in the Abstract or in the title of the paper that the characteristics of `a novel approach of assessing PAF` would be presented, the discussion would have been satisfactory.
  • As suggested.
  • Lines 253-309: Reconstruct the Discussion section completely, as follows:
    • Ensure a logical flow and comprehensiveness in the presentation of the comparison of own results with the results of similar studies by other authors (not only own previous studies) in other countries, and
    • provide an adequate explanation for possible differences in results between different studies.
    • Discuss primarily PAF for the 13 comorbidities presented in this paper.
    • 263-6
    • Give an explanation why variables that are not presented in the Results section of this paper are discussed, such as family history, aspirin, etc.
    • 275-9
    • At the end of the Discussion section, introduce the subsection `Strength and limitations of the study`, with a discussion of all limitations, potential sources of biases, etc.
    • 322-342
    • Pay special attention to the fact that in this version of the paper, the Discussion section ends with the citation of reference No. 30, and that a total of 34 references are listed in the list of References (references 31-34 are not cited in the text). Correct this.
    • These were/are included.
  • Line 309: Discuss the differences in PAF (%) of hepatobiliary cancer after comorbidities, 1997-2018, by gender, in accordance with the results presented in this paper. Provide an explanation for the differences found.
  • 300-5
  • Line 309: Discuss the differences in the distribution of population attributable fraction PAF of liver, gallbladder and bile ducts cancers for each comorbidity, by gender, in accordance with the results presented in this paper. Provide an explanation for the differences found.
  • L282-292
  • Line 309: Especially discuss the result of this paper that the female risk profile showed that obesity was associated with no female cancers.
  • 304-5  
  • Line 310: The Conclusions section correctly highlights the most important results presented in this paper, except for the sentence on Lines 318-321, which does not belong in the Conclusions section but in the Discussion section. Correct this.   
  • L314-6

We thank the reviewer!

Round 2

Reviewer 1 Report

The revised version is OK. Thank you!

Reviewer 2 Report

Thank you for the opportunity to review the manuscript (ID: rak-2439749). Overall, the authors submitted a version of the manuscript in which significant corrections were made. The revised manuscript is satisfactorily clear and informative for the topic it describes. The authors satisfactorily responded to all my comments and provided appropriate explanations.  

Thanks to the authors.   

The quality of English language is appropriate.